# Polynomial Time Learning-Augmented Algorithms for NP-hard Permutation Problems

**Evripidis Bampis** [1]  **Bruno Escoffier** [1]  **Dimitris Fotakis** [2] [3]  **Panagiotis Patsilinakos** [4]  **Michalis Xefteris** [5]

## Abstract

We consider a learning-augmented framework for NP-hard permutation problems. The algorithm has access to predictions telling, given a pair $u, v$ of elements, whether $u$ is before $v$ or not in an unknown fixed optimal solution. Building on the work of Braverman and Mossel (SODA 2008), we show that for a class of optimization problems including scheduling, network design and other graph permutation problems, these predictions allow to solve them in polynomial time with high probability, provided that predictions are true with probability at least $1/2 + \epsilon$, for any given constant $\epsilon > 0$. Moreover, this can be achieved with a parsimonious access to the predictions.

## 1. Introduction

In recent years, advancements in Machine Learning (ML) have significantly influenced progress in solving optimization problems across a wide range of fields. By leveraging historical data, ML predictors are utilized every day to tackle numerous challenges. These developments have motivated researchers in algorithms to incorporate ML predictions into algorithm design for optimization problems. This has given rise to the vastly growing field of *learning-augmented algorithms*, also known as *algorithms with predictions*. In this framework, it is assumed that predictions about a problem's input are provided by a black-box ML model. The objective is to use these predictions to develop algorithms that outperform existing ones when the predictions are sufficiently accurate.

The idea of learning-augmented algorithms was initially in-

---
*Equal contribution [1]Sorbonne Université, CNRS, LIP6, F-75005 Paris, France [2]National Technical University of Athens, Greece [3]Archimedes, Athena Research Center, Greece [4]Université Paris-Dauphine, Université PSL, CNRS, LAMSADE, 75016, Paris, France [5]Athens University of Economics and Business, Athens, Greece. Correspondence to: Michalis Xefteris <mxefteris@hotmail.com>.

*Proceedings of the $42^{nd}$ International Conference on Machine Learning*, Vancouver, Canada. PMLR 267, 2025. Copyright 2025 by the author(s).

troduced by Mahdian, Nazerzadeh, and Saberi, who applied it to the problem of allocating online advertisement space for budget-constrained advertisers (Mahdian et al., 2007). Later, Lykouris and Vassilvitskii formalized the framework, studying the online caching problem using predictions (Lykouris & Vassilvitskii, 2021). The main emphasis in the field of algorithms with predictions has been on online optimization, as predicting the future of a partially unknown input instance is a natural approach. However, in the past few years, the field has expanded into various other areas. An almost complete list of papers in the field can be found in (Lindermayr & Megow).

More relevant to this work, algorithms with predictions have been used to address NP-hard optimization problems and overcome their computational challenges. The first learning-augmented algorithms applied to NP-hard problems were focused on clustering, as seen in (Gamlath et al., 2022; Ergun et al., 2022; Nguyen et al., 2023), and other graph optimization problems (Chen et al., 2022). Moreover, several papers have studied MAXCUT with predictions, including (Bampis et al., 2024; Cohen-Addad et al., 2024; Ghoshal et al., 2024; Dong et al., 2025). Cohen-Addad et al. investigated the approximability of MAXCUT with predictions in two models (Cohen-Addad et al., 2024). In the first model, similar to the one used in this work, they assumed predictions for each vertex (on its position in an optimal cut) that are correct with probability $1/2 + \epsilon$, for a given $\epsilon > 0$, and presented a polynomial-time $(0.878 + \tilde{\Omega}(\epsilon^4))$-approximation algorithm. In the second model, they receive a correct prediction for each vertex with probability $\epsilon > 0$ (and no information otherwise) and designed a $(0.858 + \Omega(\epsilon))$-approximation algorithm. Ghoshal et al. also studied MAXCUT and MAX2-LIN in both models (Ghoshal et al., 2024). Furthermore, (Braverman et al., 2024) studied Maximum Independent Set within the framework of learning-augmented algorithms, adopting the aforementioned first model. Finally, (Antoniadis et al., 2024) studied approximation algorithms with predictions for several NP-hard optimization problems within a prediction model different from the one used in this work.

In this paper, we design learning-augmented algorithms for NP-hard optimization problems. Our approach does not use all available predictions for the problem at hand but instead utilizes the predictor selectively. This aligns with the con-

cept of parsimonious algorithms, introduced by (Im et al., 2022), which aim to limit the number of predictions used, assuming that obtaining additional predictions can be computationally expensive. Here, we consider problems whose feasible solutions can be represented as permutations. There are $n$ input elements for an optimization problem denoted by $a_1, \ldots, a_n$. A permutation (ordering) $\sigma$ corresponds to the solution $(a_{\sigma(1)}, \ldots, a_{\sigma(n)})$.

Regarding the prediction model, we adopt the following probabilistic framework. For each pair $i, j$ we can get a prediction query $q(a_i, a_j)$ that denotes whether $a_i$ precedes $a_j$ or not in a fixed optimal solution (permutation). Each prediction is independently correct with probability at least $1/2 + \epsilon$, for a given constant $\epsilon > 0$. An algorithm has access to $\binom{n}{2}$ predictions. A formal description of the model is given in Section 2.

In this work, we use these prediction queries to solve NP-hard optimization problems exactly with high probability. We design a novel framework which is capable of handling permutation optimization problems that exhibit one of two key properties: the *decomposition* property and the *c-locality* property in their objective function (see Section 3 for formal definitions).

The decomposition property states that solving a subproblem $\mathcal{I}(i, j)$ (between positions $i$ and $j$ in the permutation) of the optimization problem at hand optimally depends only on the set of elements in positions in $[i, j]$, the permutation $\sigma(i, j)$ of these elements in $[i, j]$, and the set of elements to the left of $i$ and to the right of $j$, but not on their order. On the other hand, the $c$-locality property states that the cost function of the problem depends only locally (with respect to the permutation) on pairs of distinct elements.

More specifically, we adjust and extend the approach of Braverman and Mossel (Braverman & Mossel, 2008; 2009) for the problem of sorting from noisy information and give the following theorem for a family of optimization problems (see Section 4 for its proof).

**Theorem 1.1.** *If the objective function of a permutation optimization problem $P$ either exhibits the decomposition property or is $c$-local, then $P$ can be solved exactly with high probability in polynomial time, using $O(n \log n)$ prediction queries.*

Therefore, if a permutation optimization problem is either decomposable or $c$-local, it can be solved with high probability in polynomial time within our prediction-based framework. We note that the running time depends on the accuracy of the prediction queries, i.e., on $\epsilon$. To illustrate these properties, we examine several example problems: Maximum Acyclic Subgraph, Minimum Linear Arrangement, a scheduling problem as examples of decomposable problems, the Traveling Salesperson Problem (TSP) and social

welfare maximization in keyword auctions with externalities and window size as representatives of $c$-local problems. All these are well-known NP-hard problems and cannot be solved exactly in polynomial time without predictions unless $\mathcal{P} = \mathcal{NP}$ (deterministic), or $\mathcal{NP} \subseteq \mathcal{BPP}$ (randomized). Moreover, the framework can naturally be extended to address a variety of other NP-hard problems with similar structural properties.

Another important aspect of our framework is that it does not query all possible pairs but instead makes only $O(n \log n)$ queries, making it parsimonious with respect to the number of predictions used. We note that this is a tight bound on the number of queries, as $\Omega(n \log n)$ queries are necessary even in the case of perfect predictions already for the Noisy Sorting Without Resampling problem (see Section 2.1). Indeed, it becomes sorting by comparisons (where comparisons are queries), for which it is well known that $\Omega(n \log n)$ comparisons are needed (even for randomized algorithms).

The proof of Theorem 1.1 demonstrates that, for the permutation problems under consideration, knowing an approximation of each $\sigma^*(i)$ (in an optimal solution $\sigma^*$) within an additive $O(\log n)$ bound is sufficient to solve the problem in polynomial time. In Section 5, we first show that this $O(\log n)$ approximation is not always sufficient for polynomial time solvability. Finally, we prove that the $O(\log n)$ bound is tight, as there are decomposable and $c$-local problems where an additive approximation of $f(n) \log n$ is not enough to solve these problems in polynomial time, for any unbounded function $f$.

We conclude this section by a brief discussion on the prediction model. Similarly as several recent articles on the topic, our framework requires good predictions of optimal solutions for hard optimization problems, which is a strong assumption. Whether ML algorithms are or will soon be able to provide such good predictions or not is currently an open question, and a large number of recent works do focus on getting predictions for discrete optimization problems (see (Bengio et al., 2021; Cappart et al., 2023) for survey-like papers, and (Khalil et al., 2017) for a specific work on predictions for problems including (Euclidean) TSP). This recent but fast-growing field makes it reasonable to hope for some accurate predictors for some permutation problems in the near future. We note also that there are restricted settings where one could outline how such noisy predictions of the optimal permutation can be learned. E.g., consider a setting where instances are stationary and instance stationarity implies that the optimal solution / permutation of each instance is a noisy sample from a Mallows distribution $\mathcal{M}(\pi^*, \beta)$ with $\pi^*$ corresponding to the optimal permutation of the "mean" instance and the noise parameter $\beta$ accounts for the variance of the instance distribution. Such a Mallows dis-

tribution $\mathcal{M}(\pi^*, \beta)$ can be learned from instances sampled independently from the instance distribution (Busa-Fekete et al., 2019). Samples from $\mathcal{M}(\pi^*, \beta)$ can be used as predictions in the form of noisy rankings. For fixed $\beta$, this provides whp a sufficiently good approximation of the ranking (up to an additive $O(\log n)$ term on the position of each element) so that our approach applies (see (Braverman & Mossel, 2009), where this case is discussed).

## 2. Background and Overview

### 2.1. Definitions

Formally, we consider the following probabilistic prediction model, which is inspired by the noisy query model studied in (Braverman & Mossel, 2008).

**Definition 2.1.** Let $A = \{a_1, \ldots, a_n\}$ and $\sigma^*$ be a permutation from $[1, n]$ to $[1, n]$. For each pair $(a_\ell, a_t)$ in $\binom{A}{2}$ the result of a prediction query for $(a_\ell, a_t)$, with respect to $\sigma^*$, is $q(a_\ell, a_t) \in \{-1, 1\}$ where $q(a_\ell, a_t) = -q(a_t, a_\ell)$. We assume that:

- for each $1 \leq i < j \leq n$ the probability that $q(a_{\sigma^*(i)}, a_{\sigma^*(j)}) = 1$ is at least $\frac{1}{2} + \epsilon$, $0 < \epsilon < 1/2$,

- the queries $\{q(a_\ell, a_t) : 1 \leq \ell < t \leq n\}$ are independent conditioned on $\sigma^*$.

In this definition, the query $q(a_\ell, a_t)$ asks whether $a_\ell$ precedes $a_t$ in $(a_{\sigma^*(1)}, \ldots, a_{\sigma^*(n)})$ or not. The first item states that the prediction is correct with probability at least $1/2 + \epsilon$.

Given the prediction queries, we are interested in finding a permutation that maximizes the number of agreements with the queries. Formally:

**Definition 2.2.** Given $\binom{n}{2}$ prediction queries $q(a_\ell, a_t)$, the score $s_q(\pi)$ of a permutation $\pi : [1, n] \to [1, n]$ is given by

$$s_q(\pi) = \sum_{i < j} q(a_{\pi(i)}, a_{\pi(j)}). \tag{1}$$

We say that a permutation $\pi^*$ is s-optimal if $\pi^*$ is a maximizer of (1) among all permutations.

The *Noisy Sorting Without Resampling (NSWR)* problem, defined in (Braverman & Mossel, 2008), is the problem of finding an s-optimal permutation $\pi^*$ with respect to a (hidden) permutation $\sigma^*$ assuming that $q$ satisfies Definition 2.1 with $p = 1/2 + \epsilon$, $\epsilon > 0$.

As mentioned in the introduction, we consider in this work permutation problems, i.e., optimization problems whose solutions of an instance $I$ are permutations of $n$ elements of a set $A$ of $I$ (vertices or edges in a graph, jobs in a scheduling problem,...). So the goal is to maximize or minimize $f_I(\sigma)$ for $\sigma : [1, n] \to [1, n]$. Here, $a_{\sigma(i)} \in A$ is

the element of $A$ that is in position $i$ in the permutation (i.e., the permutation is $(a_{\sigma(1)}, a_{\sigma(2)}, \ldots, a_{\sigma(n)})$). To deal with feasibility constraints, $f_I(\sigma) = \infty$ if $\sigma$ is unfeasible (for a minimization problem, $-\infty$ for a maximization problem).

A core part of our work will focus on permutation problems for which we have an additional information, which is an approximation of $\sigma^*(i)$ for an optimal solution $\sigma^*$. We formalize this in the following definition.

**Definition 2.3.** Given a permutation problem $P$ and a permutation $(a_1, a_2, \ldots, a_n)$, $P$ is *k-position enhanced* if we know that there exists an optimal solution $\sigma^*$ such that for all $i$, $|\sigma^*(i) - i| \leq k$.

### 2.2. Framework Overview

The NSWR problem has been introduced and studied in (Braverman & Mossel, 2008). They showed the following result.

**Theorem 2.4.** *(Braverman & Mossel, 2008) There exists a randomized algorithm that for any $\alpha > 0$ finds an optimal solution to NSWR (s-optimal) with $p = 1/2 + \epsilon, \epsilon > 0$ in time $n^{O((\alpha+1)\epsilon^{-4})}$ except with probability $n^{-\alpha}$. Moreover, the algorithm asks $O(n \log n)$ queries.*

The proof of this theorem mainly relies on two results. The first one shows that with high probability an optimal solution $\pi^*$ of NSWR (s-optimal) is close to the "hidden" permutation $\sigma^*$.

**Theorem 2.5.** *(Braverman & Mossel, 2008) Consider the NSWR problem, with $p = 1/2 + \epsilon, \epsilon > 0$, with respect to a permutation $\sigma^*$ and let $\pi^*$ be any s-optimal order assuming that $q$ satisfies Definition 2.1. Let $\alpha > 0$. Then there exists a constant $c(\alpha, \epsilon)$ such that except with probability $O(n^{-\alpha})$ it holds that*

$$\max_i |\sigma^*(i) - \pi^*(i)| \leq c \cdot \log n = O(\log n).$$

The second result is a dynamic programming (DP) algorithm showing that $k$-position enhanced NSWR is solvable in $O(2^{O(k)} n^2)$. Then a specific iterative procedure allows for the computation of an optimal solution of NSWR. Very roughly speaking, the use of queries allows for a $k$-position enhancement for NSWR with $k = O(\log n)$ (thanks to Theorem 2.5), and then the DP algorithm works in polynomial time $O(2^{O(k)} n^2) = n^{O(1)}$.

In this work, we build upon these results to tackle various problems in our prediction setting. Roughly speaking, our framework first generates a warm-start solution using the prediction queries and then utilizes this solution to solve the problem with dynamic programming. The idea of leveraging predictions to obtain a warm-start solution has been explored in a series of papers in the literature (Dinitz et al., 2021; Sakaue & Oki, 2022). The following lemma, which makes a

connection with the aforementioned results on NSWR, will allow us to get the polynomial time algorithms claimed in Theorem 1.1.

**Lemma 2.6.** *Suppose that a $k$-position enhanced version of permutation problem $P$ is solvable in polynomial time for $k = O(\log n)$. Then $P$ can be solved exactly in polynomial time with high probability, using $O(n \log n)$ prediction queries.*

*Proof.* Let $\sigma^*$ be an optimal permutation for an instance of the optimization problem $P$. By making $O(n \log n)$ queries, according to Theorem 2.4 we can get in polynomial time with high probability an optimal solution $\pi^*$ for the NSWR problem relative to $\sigma^*$.

From Theorem 2.5, we know that with high probability $|\sigma^*(i) - \pi^*(i)| = O(\log n)$ for all $i$. Equivalently, for all $j$:

$$|\sigma^* \circ \pi^{*-1}(j) - j| = O(\log n).$$

As by assumption the $k$-position enhanced version of $P$ is solvable in polynomial time for $k = O(\log n)$, we can find $\sigma^* \circ \pi^{*-1}$, hence $\sigma^*$, in polynomial time. $\square$

Motivated by Lemma 2.6, we exhibit two sufficient conditions for a permutation problem to be polynomial time solvable when being $k$-positioned enhanced, for $k = O(\log n)$. These conditions (decomposability and $c$-locality) and their illustration on classical optimization problems are given in Section 3. The proofs that under these conditions $O(\log n)$-positioned enhanced permutation problems are polynomial time solvable are in Section 4. They are based on DP algorithms, one of which being a generalization of the one of (Braverman & Mossel, 2008).

## 3. Decomposability and $c$-locality

We now give the definitions for the decomposition property and $c$-locality, and illustrate them with some problems expressible in these ways. As explained before, each of these properties will allow to design DP algorithms, which is the key step to derive Theorem 1.1.

### 3.1. Decomposition property

To design DP algorithms, we will consider subproblems. Intuitively, for $i < j$ we will consider subproblems of finding and ordering elements in positions $i$ to $j$, i.e., $(a_{\sigma(i)}, \ldots, a_{\sigma(j)})$, and need a recurrence that allows expressing the subproblem between $i$ and $j$ as a combination of the subproblems between $i$ and $s$, and between $s + 1$ and $j$ (for $i < s < j$). A difficulty is that finding and ordering elements from positions $i$ to $j$ typically depends on the elements before (between positions 1 and $i - 1$) and after (between positions $j + 1$ and $n$), and on their respective

ordering. Roughly speaking, our decomposition property holds when finding and ordering elements from positions $i$ to $j$ only depends on the *set* of elements before $i$ and after $j$, not on their particular ordering.

We now formalize this idea, and illustrate it on three different problems. For a permutation $\sigma : [1, n] \to [1, n]$, we denote $\sigma(i, j)$ the subpermutation of $\sigma$ on $[i, j]$, $S_\sigma(i, j)$ the set $\{\sigma(i), \ldots, \sigma(j)\}$ (indices in positions between $i$ and $j$), $L_\sigma(k) = S_\sigma(1, k)$ (first $k$ indices, $L$ stands for left) and $R_\sigma(k) = S_\sigma(k, n)$ (indices in position between $k$ and $n$, $R$ stands for right).

**Definition 3.1.** Let $P$ be a permutation problem, with objective function $f$. We say that $P$ fulfills the decomposition property if there exists a function $g$ such that:

- $f_I(\sigma) = g(1, n, \emptyset, \emptyset, \sigma)$;

- For any $i < s < j$ in $\{1, \ldots, n\}$ and permutation $\sigma$:

$$g(i, j, L_\sigma(i-1), R_\sigma(j+1), \sigma(i,j)) =$$
$$g(i, s, L_\sigma(i-1), R_\sigma(s+1), \sigma(i,s))$$
$$+ g(s+1, j, L_\sigma(s), R_\sigma(j+1), \sigma(s+1,j))$$
$$+ h(L_\sigma(i-1), R_\sigma(j+1), S_\sigma(i,s), S_\sigma(s+1,j)),$$

for some function $h$.

Note that $h$ does not depend on the permutation of any elements, only on sets of elements. In this definition, we express the objective function on subproblem $(i, j)$ by a function $g$ that depends on the subpermutation $\sigma(i, j)$ between $i$ and $j$ and on sets (not positions) of elements before $i$ and after $j$ ($L_\sigma(i - 1)$ and $R_\sigma(j + 1)$).

The decomposition property states that the value is the sum of the value on the subproblem $(i, s)$ (referred to as the "left call" in the sequel), the value on the subproblem $(s + 1, j)$ (referred to as the "right call"), and a quantity (function $h$) that depends only on the sets (not the ordered sets) of elements involved in the decomposition.

**Maximum Acyclic Subgraph.** In the Maximum Acyclic Subgraph problem, we are given a simple directed graph $G = (V, E)$. The goal is to find a permutation $\sigma : [1, n] \to [1, n]$ which maximizes $f(\sigma) = |\{(v_{\sigma(i)}, v_{\sigma(j)}) \in E : i < j\}|$. This is a well known NP-hard problem (Karp, 1972).

As this problem only depends on the relative order of elements (endpoints of arcs) and not on their exact position, it is easy to express it in the form of Definition 3.1. For given $i < j$ and $\sigma(i, j)$, we simply define:

$$g(i, j, L, R, \sigma(i,j)) = |\{(v_{\sigma(\ell)}, v_{\sigma(r)}) \in E : i \leq \ell < r \leq j\}|.$$

This is just the number of arcs "well ordered" by $\sigma$ with both endpoints between positions $i$ and $j$ ($g$ is independent

of $L$ and $R$). Obviously the first item of Definition 3.1 is verified (for $i = 1$ and $j = n$ we count the total number of arcs "well ordered" by $\sigma$).

For the second item, $g(i, s, L_\sigma(i-1), R_\sigma(s+1), \sigma(i, s))$ (resp. $g(s+1, j, L_\sigma(s), R_\sigma(j+1), \sigma(s+1, j)))$ counts the number of arcs well ordered by $\sigma$ with both endpoints in positions between $i$ and $s$ (resp. between $s+1$ and $j$). So, to get $g(i, j, L_\sigma(i-1), R_\sigma(j+1), \sigma(i, j))$, we only have to add arcs $(v_{\sigma(\ell)}, v_{\sigma(r)})$ with $i \leq \ell \leq s$ and $s+1 \leq r \leq j$. In other words, if we denote $cut(A, B)$ the number of arcs $(v_\ell, v_r)$ with $\ell \in A$ and $r \in B$, we have

$$h(L_\sigma(i-1), R_\sigma(j+1), S_\sigma(i, s), S_\sigma(s+1, j) = cut(S_\sigma(i, s), S_\sigma(s+1, j)).$$

Note that this immediately generalizes to any problem whose objective function only depends on the relative order of endpoints of arcs (for instance the weighted version of Maximum Acyclic Subgraph), not on their exact positions.

**Minimum Linear Arrangement.**   Let $G = (V, E)$ be a simple undirected graph. Given a permutation $\sigma : [1, n] \rightarrow [1, n]$, let us call the weight of an edge as the absolute difference between the positions assigned to its endpoints in $\sigma$. Minimum Linear Arrangement is a well known NP-hard (Garey & Johnson, 1979) problem which consists of finding a permutation of the vertices of $G$ such that the sum of the weights of its edges is minimized. More formally, we would like to find a permutation $\sigma : [1, n] \rightarrow [1, n]$ that minimizes $\sum_{\{v_{\sigma(i)}, v_{\sigma(j)}\} \in E} |j - i|$.

While Maximum Acyclic Subgraph deals (only) with the order of endpoints of arcs in the permutation, Minimum Linear Arrangement depends on their exact position. We now show that the decomposition property also allows to deal with such a problem. The idea is the following: consider $i < j$, and fix the subpermutation $\sigma(i, j)$, the sets $L_\sigma(i-1)$ of elements "on the left", and $R_\sigma(j+1)$ of elements "on the right". Then (see Figure 1 for an illustration of the contribution of edges):

- for an edge with both endpoints in $S_\sigma(i, j)$, we know its exact contribution (as $\sigma(i, j)$ is fixed). These edges have a global contribution $N_1 = \sum_{\{v_{\sigma(\ell)}, v_{\sigma(r)}\} \in E, i \leq \ell, r \leq j} |r - \ell|$. Note that $N_1$ depends only on $\sigma(i, j)$.

- For an edge with one endpoint in the left part $L_\sigma(i-1)$ and one endpoint in $S_\sigma(i, j)$, i.e., $\{v_{\sigma(\ell)}, v_{\sigma(r)}\}$ with $\ell < i$ and $i \leq r \leq j$, we will charge only for this edge the (yet partial) contribution $r - i$ (instead of $r - \ell$, as the exact left position is not fixed yet). Let $N_2$ be the sum of these contributions (note that $N_2$ depends only on $L_\sigma(i-1)$ and $\sigma(i, j)$).

- Similarly, for an edge with one endpoint in the right part $R_\sigma(j+1)$ and one endpoint in $S_\sigma(i, j)$, i.e., $\{v_{\sigma(\ell)}, v_{\sigma(r)}\}$ with $i \leq \ell \leq j$ and $j < r$, we will charge only for this edge the (yet partial) contribution $j - \ell$. Let $N_3$ be the sum of these contributions (note that $N_3$ depends only on $R_\sigma(j+1)$ and $\sigma(i, j)$).

We then define $g(i, j, L_\sigma(i-1), R_\sigma(j+1), \sigma(i, j)) = N_1 + N_2 + N_3$.

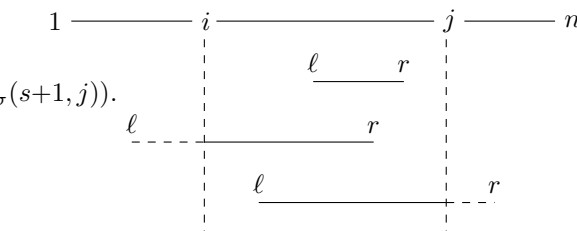

*Figure 1.* Contribution in $g$ of edges, depending on the positions of their extremities. The contribution is in solid line. If $\ell < i$ and $r > j$ the contribution is 0.

It is clear that the first item of Definition 3.1 is satisfied (when $i = 1$ and $j = n$, $N_1$ equals the objective function, $N_2 = N_3 = 0$).

For the second item, let $i < s < j$. Consider an edge $\{v_{\sigma(\ell)}, v_{\sigma(r)}\}$. We show, for each possible case, how to recover the contribution of this edge to subproblem $(i, j)$ (i.e., in $g(i, j, L_\sigma(i-1), R_\sigma(j+1), \sigma(i, j)))$, see Figure 2 for an illustration:

- If both $\ell$ and $r$ are in $[i, s]$, its contribution in $g(i, j, L_\sigma(i-1), R_\sigma(j+1), \sigma(i, j))$ is already counted in the "left call" $g(i, s, L_\sigma(i-1), R_\sigma(s+1), \sigma(i, s))$.

- Similarly, if both $\ell$ and $r$ are in $[s+1, j]$, its contribution in $g(i, j, L_\sigma(i-1), R_\sigma(j+1), \sigma(i, j))$ is already counted in the "right call" $g(s+1, j, L_\sigma(s), R_\sigma(j+1), \sigma(s+1, j))$.

- If $i \leq \ell \leq s$ and $s+1 \leq r \leq j$, then the contribution is $(s - \ell)$ in the left call and $r - (s+1)$ in the right call, so in total $r - \ell - 1$. It only missed 1 to get the correct contribution $r - \ell$. So for these edges we have to add in total a contribution $C_1 = cut(S_\sigma(i, s), S_\sigma(s+1, j))$.

- If $\ell < i$ and $i \leq r \leq s$, then the contribution on the left call is $r - i$, the correct one. Similarly, if $s+1 \leq \ell \leq j$ and $j + 1 \leq r$ the contribution of the right call is the correct one $(j - \ell)$.

- If $\ell < i$ and $s+1 \leq r \leq j$, the contribution in the left call is 0, the one in the right call is $r - (s+1)$. It misses $s + 1 - i$ to get the correct contribution $r - i$.

So for these edges we have to add in total $C_2 = (s + 1 - i) \cdot cut(L_\sigma(i-1), S_\sigma(s+1, j))$.

- Similarly, if $i \leq \ell \leq s$ and $r > j$, to get the correct charge we miss $C_3 = (j - s) \cdot cut(R_\sigma(j+1), S_\sigma(i, s))$.

Then, we define $h(L_\sigma(i-1), R_\sigma(j+1), S_\sigma(i, s), S_\sigma(s+1, j)) = C_1 + C_2 + C_3$.

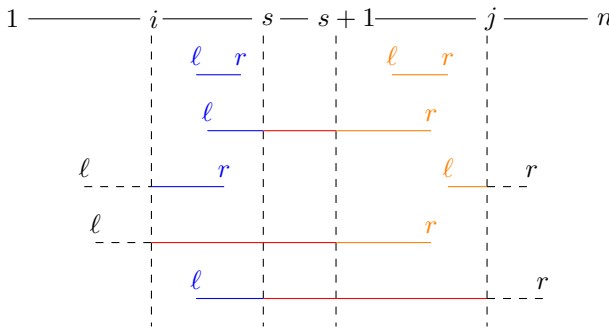

*Figure 2.* Decomposition in $g$: blue (resp. orange) corresponds to the contribution in the left call (resp. right call), red corresponds to missing contributions (that are counted in $h$). If $\ell < i$ and $r > j$ the contribution is 0.

**Single-machine Sum of Completion Time Problem.** We consider the following scheduling problem (denoted $1|prec| \sum C_j$ in the classical Graham's notation for scheduling problems), known to be strongly NP-hard (Lawler, 1978): we are given a set $J$ of $n$ jobs and a set of precedence constraints forming a DAG $(J, A)$. The goal is to order the jobs, respecting precedence constraints, so as to minimize the sum of completion time of jobs.

Consider $i < j$, $\sigma(i, j)$ and $L_\sigma(i-1)$. With this information, we can compute the completion time of each job in $S_\sigma(i, j)$ (as we know the jobs before them ($L_\sigma(i - 1)$) and the order $\sigma(i, j)$ of jobs in $S_\sigma(i, j)$). So we simply define $g$ as the sum of these completion times ($g$ depends on $S_\sigma(i, j)$ and $L_\sigma(i - 1)$), with of course value $\infty$ if the order $\sigma(i, j)$ violates any constraints.

Item 1 of Definition 3.1 is trivially satisfied. For item 2, the left call computes the sum of completion times for jobs in $S_\sigma(i, s)$, and the right call the sum of completion times for jobs in $S_\sigma(s + 1, j)$. Then $h$ is $\infty$ if a constraint is violated (between a job whose position is between $s + 1$ and $j$, and a job whose position is between $i$ and $s$), and 0 otherwise.

Here again, this generalizes to many other single machine scheduling problems (involving other constraints and/or weights on jobs and/or other objective functions like sum of tardiness).

### 3.2. $c$-locality

The $c$-locality property states that the cost function of the problem depends only locally (with respect to the permutation/solution) on pairs of distinct elements. More formally, we have the following definition.

**Definition 3.2.** Let $P$ be a permutation problem, with cost function $f$. We say that $P$ is $c$-local if, on an instance $I$ asking for a permutation $\sigma$ on a set $A = \{a_1, \ldots, a_n\}$:

$$f_I(\sigma) = \sum_i cost_I(a_{\sigma(i-c)}, a_{\sigma(i-c+1)}, \ldots, a_{\sigma(i)}).$$

for some cost function[1] $cost_I$.

**TSP.** Given a complete graph on set $V$ of vertices and a distance function $d(v_i, v_j)$ on pair of vertices, TSP asks to find a permutation $\sigma$ that minimizes $\sum_{i=1}^n d(v_{\sigma(i)}, v_{\sigma(i+1)})$ (where $v_{\sigma(n+1)}$ is $v_{\sigma(1)}$). This NP-hard problem (Garey & Johnson, 1979) is then trivially 1-local (we can easily reformulate to get rid of the last term $d(v_{\sigma(n)}, v_{\sigma(1)})$).

Other routing problems can be shown to be 1-local as well.

**Social Welfare Maximization in Keyword Auctions with Externalities.** In standard Keyword Auctions (Edelman et al., 2007; Varian, 2007), a set $N = \{1, \ldots, n\}$ of advertisers compete over a set $K = \{1, \ldots, k\}$ of advertisement (or simply ad) slots, where $k \leq n$. Each player $i \in N$ has a valuation $v_i$ per click and their ad has an intrinsic click probability $q_i \in (0, 1]$. Each slot $j \in K$ is associated with a click-through rate (ctr) $\lambda_j$, with $1 \geq \lambda_1 > \lambda_2 > \cdots \lambda_k > 0$. The overall ctr of an ad $i$ in slot $j$ is $\lambda_j q_i$. In the following, we assume that $k = n$, i.e., the number of slots $k$ is equal to the number of ads $n$, for simplicity, by setting $\lambda_{k+1} = \cdots = \lambda_n = 0$ in case where $k < n$.

We aim to compute an assignment $\pi : N \to K$ of ads to slots (which for $k = n$ is a permutation of ads) so that the resulting expected *social welfare*, which is $\sum_{i=1}^n v_i \lambda_{\pi(i)} q_i$, is maximized.

In Keyword Auctions with Externalities (Gatti et al., 2018; Fotakis et al., 2011), the overall click-through rate of an ad $i$ appearing in slot $\pi(i)$ also depends on the ads appearing in the $c$ slots above $\pi(i) - c, \pi(i) - c + 1, \ldots, \pi(i) - 1$, where $c$ is known as the *window size* and quantifies the scope of users' attention and memory when they process the ad list (Athey & Ellison, 2011). We let $Q_i(\pi)$ denote the $i$'s ctr given the influence of the $c$ ads above $i$ in $\pi$. Then, we aim to compute an assignment $\pi : N \to [n]$ of ads to slots so that the resulting expected social welfare under externalities, which is $\sum_{i=1}^n v_i \lambda_{\pi(i)} Q_i(\pi)$, is maximized.

Social welfare maximization in keyword auctions with ex-

---

[1]To be more precise, to deal with the cases $i \leq c$ in the sum it should be $cost_I(a_{\sigma(t)}, a_{\sigma(t+1)}, \ldots, a_{\sigma(i)})$ for $t = \max\{i-c, 1\}$.

ternalities is NP-hard (Fotakis et al., 2011; Gatti et al., 2018) and is an example of a $c$-local permutation problem.

## 4. Proof of Theorem 1.1

In this section, we present the proof of Theorem 1.1, the main result of this paper. Using Lemma 2.6, what remains to be shown is that if a problem is decomposable or $c$-local, then it is polynomial time solvable when being $O(\log n)$-position enhanced. We provide a corresponding DP algorithm for decomposition in Lemma 4.1 (which extends the one of (Braverman & Mossel, 2008)) and for $c$-locality in Lemma 4.2

**Lemma 4.1.** *If a permutation optimization problem $P$ is decomposable, then its $k$-position enhanced version is solvable in time $O(n \cdot 2^{O(k)} \cdot t)$ where $t$ is the time to compute the value of a solution.*

This gives a polynomial computation time when $k = O(\log n)$, provided that $t \in poly(n)$.

*Proof.* We will use dynamic programming to find an optimal solution for the optimization problem $P$. Let $\sigma^*$ be an (unknown) optimal permutation such that $|\sigma^*(i) - i| \leq c \log n$ for all $i$.

Let $i < j$ be any indices. Let $I^*(i, j)$ denote the elements in positions between $i$ and $j$ in $\sigma$, i.e.,

$$I^*(i, j) = \{a_{\sigma(i)}, a_{\sigma(i+1)}, \dots, a_{\sigma(j)}\}.$$

Moreover, let

$$I_L^*(i) = \{a_{\sigma(1)}, a_{\sigma(2)}, \dots, a_{\sigma(i-1)}\}$$

be the elements of the left of $i$ and

$$I_R^*(j) = \{a_{\sigma(j+1)}, a_{\sigma(j+2)}, \dots, a_{\sigma(n)}\}$$

the elements on the right of $j$. By assumption, we have $I_L^-(i) \subseteq I_L^*(i) \subseteq I_L^+(i)$, $I^-(i, j) \subseteq I^*(i, j) \subseteq I^+(i, j)$ and $I_R^-(j) \subseteq I_R^*(j) \subseteq I_R^+(j)$ where

$$I_L^+(i) = \{a_1, a_2, \dots, a_{i+k-1}\},$$

$$I_L^-(i) = \{a_1, a_2, \dots, a_{i-k-1}\},$$

$$I^+(i, j) = \{a_{i-k}, a_{i-k+1}, \dots, a_{j+k}\},$$

$$I^-(i, j) = \{a_{i+k}, a_{i+k+1}, \dots, a_{j-k}\},$$

$$I_R^+(j) = \{a_{j+1-k}, a_{j+2-k}, \dots, a_n\},$$

$$I_R^-(j) = \{a_{j+1+k}, a_{j+2+k}, \dots, a_n\}.$$

Thus, selecting the sets $I_L^*(i)$, $I^*(i, j)$ and $I_R^*(j)$ involves selecting $k$ elements from a list of $2k$ elements for $I_L^*(i)$ and $2k$ elements from $4k$ elements for $I^*(i, j)$. Once $I_L^*(i)$,

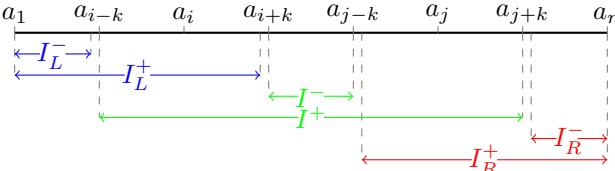

*Figure 3.* Selecting the set $I^*$ involves choosing $j - i + 1$ elements that include all elements of $I^-$ and are contained within $I^+$ (highlighted in green). This corresponds to selecting $2k$ elements from the list $\{a_{i-k}, a_{i-k+1}, \dots, a_{i+k-1}, a_{j-k+1}, \dots, a_{j+k}\}$ of $4k$ elements. Similarly, selecting $I_L^*$ requires choosing $i - 1$ elements that include $I_L^-$ and are contained within $I_L^+$ (in blue). This involves selecting $k$ elements from the list $\{a_{i-k}, \dots, a_{i+k-1}\}$ of $2k$ elements. The same logic applies for $I_R^*$ (in red).

$I^*(i, j)$ are fixed, the remaining elements belong to $I_R^*(j)$ (see Figure 3 for an illustration). Thus the number of different guesses we have to do is bounded by $2^{2k+4k} = 2^{6k}$.

Now, let us define for any $i, j$ the set $\mathcal{S}(i, j)$ of sets $I'(i, j)$ of size $j - i + 1$ such that $I^-(i, j) \subseteq I'(i, j) \subseteq I^+(i, j)$. We have that $I^*(i, j) \in \mathcal{S}(i, j)$.

Moreover, let $\mathcal{L}(i - 1)$ be the set of sets $I_L'(i)$ of size $i - 1$ such that $I_L^-(i) \subseteq I_L'(i) \subseteq I_L^+(i)$. We have that $I_L^*(i) \in \mathcal{L}(i-1)$. Similarly, we define the set $\mathcal{R}(j + 1)$ of sets $I_R'(j)$ of size $n - j$.

We now define the following problem $\mathcal{I}(i, j)$: for each $I'(i, j) \in \mathcal{S}(i, j)$, each $I_L'(i) \in \mathcal{L}(i - 1)$ and each $I_R'(j) \in \mathcal{R}(j + 1)$ such that $I'(i, j)$, $I_L'(i)$ and $I_R'(j)$ are pairwise disjoint, find a permutation $\sigma'$ of elements in $I'(i, j)$ such that:

- $|\sigma'(t) - t| \leq k$,

- $\{a_{\sigma'(i)}, \dots, a_{\sigma'(j)}\} = I'(i, j)$

- $\sigma'(i, j)$ optimizes the function $g$ given in the decomposition property of $P$.

Based on the decomposition property of $g$, we will solve the problem via dynamic programming. Note that we find an optimal solution by solving $\mathcal{I}(1, n)$. Assume for the sake of simplicity that $n$ is a power of two. We will solve $\mathcal{I}(1, n)$ using the solutions of $\mathcal{I}(1, n/2)$ and $\mathcal{I}(n/2 + 1, n)$, and so on. We solve the leaves of the tree (containing only one element) in constant time, and we have it total $n - 1$ subproblems.

Let us explain how we solve $\mathcal{I}(i, j)$ using the solutions of $\mathcal{I}(i, s)$ and $\mathcal{I}(s + 1, j)$, where $i \leq s \leq j$.

Let $I'(i, j) \in \mathcal{S}(i, j)$, $I_L'(i) \in \mathcal{L}(i - 1)$ and $I_R'(j) \in \mathcal{R}(j + 1)$. Thanks to the decomposition property on $g$

(see Definition 3.1), $I'(i,j)$, $I'_L(i)$ and $I'_R(j)$ being fixed, an optimal $\sigma'(i,j)$ is composed of an optimal on $[i,s]$ and an optimal solution on $[s+1,j]$. There are at most $2^{O(k)}$ choices to partition $I'(i,j)$ into two sets $I'(i,i+s) \in \mathcal{S}(i,s)$ and $I'(s+1,j) \in \mathcal{S}(s+1,j)$. So we can compute an optimal solution of our subproblem using $2^{O(k)}$ calls to the DP table (on subproblems in $\mathcal{I}(i,s)$ and $\mathcal{I}(s+1,j)$). $\qquad\square$

Now we can also prove a similar lemma for the case of a $c$-local permutation optimization problem, using a different DP.

**Lemma 4.2.** *Let $P$ be a $c$-local permutation problem. Its $k$-position enhanced version is solvable in time $O(n \cdot 2^{2k} \cdot k^{c+1})$.*

*Proof.* We will use again dynamic programming to find an optimal solution for problem $P$. Let $i$ be any index. Let $I^*(i)$ denote the elements in an optimal order in positions between 1 and $i$ in $\sigma$, i.e., $I^*(i) = \{a_{\sigma(1)}, \ldots, a_{\sigma(i)}\}$.

By assumption, we have that $I^-(i) \subseteq I^*(i) \subseteq I^+(i)$, where

$$I^-(i) = \{a_1, a_2, \ldots, a_{i-k}\}, \quad I^+(i) = \{a_1, a_2, \ldots, a_{i+k}\}.$$

Thus, selecting the set $I^*(i)$ involves selecting $k$ elements from a list of $2k$ elements. Therefore, the number of different guesses we have to do is bounded by $2^{2k}$.

Let us now define for any $i$ the set $S_\sigma(i)$ of sets $I'(i)$ of size $i$ such that $I^-(i) \subseteq I'(i) \subseteq I^+(i)$. We have that $I^*(i) \in S_\sigma(i)$.

Similarly, we let $I^*(i+1, i+c)$ denote the elements in positions between $i+1$ and $i+c$ in $\sigma$. Here, we have to select each of these $c$ elements among a list of $2k+1$ elements, so the number of different guesses is bounded by $O(k^c)$ (as $c$ is a constant). Moreover, for each guess we also consider all different permutations of the $c$ elements (denoted by $\sigma(i, i+c)$) which takes constant time (as $c$ is constant).

We also define for any $i$ the set $\mathcal{S}(i+1, i+c)$ of sets $I'(i+1, i+c)$ of size $c$ such that $I^-(i+1, i+c) \subseteq I'(i+1, i+c) \subseteq I^+(i+1, i+c)$. We have that $I^*(i+1, i+c) \in \mathcal{S}(i+1, i+c)$.

Let us now explain how we solve the problem using dynamic programming which is based on the property of $c$-locality. For every entry of the DP table we have the following:

$$DP(S_\sigma(i), a_{\sigma(i+1)}, \ldots, a_{\sigma(i+c)}) =$$
$$\min_{\sigma(i)} \big\{ DP(S_\sigma(i) \setminus \{\sigma(i)\}, a_{\sigma(i)}, \ldots, a_{\sigma(i+c-1)})$$
$$+ cost_I(a_{\sigma(i)}, a_{\sigma(i+1)}, \ldots, a_{\sigma(i+c)}) \big\}.$$

There are at most $2k+1$ choices for $a_{\sigma(i)}$, so each DP entry can be computed in time $O(k)$. Overall, we have that the running time is $O(n \cdot 2^{2k} \cdot k^{c+1})$. $\qquad\square$

## 5. About position-enhanced permutation problems

The dynamic programming algorithms of Lemmas 4.1 and 4.2 show that $O(\log n)$-position enhanced versions of decomposable or $c$-local permutation problems are solvable in polynomial time. In this section, we show two complementary results on position-enhanced versions of permutation problems that, though technically quite easy, enlight interesting limitations to Lemma 2.6:

- First we show that there are some permutation problems which remain hard to solve in polynomial time even when being $O(\log n)$-position enhanced.

- Second, we show that this bound $O(\log n)$ is tight, meaning that there are decomposable problems and $c$-local problems which remain hard when being $f(n) \log n$-position enhanced, as soon as $f$ is unbounded.

### 5.1. A $\log(n)$-position-enhanced hard problem

Let us consider the following problem called Permutation Clique: we are given a graph $G = (V, E)$ on $n = t^2$ vertices $v_{i,j}$, $i, j = 1, \ldots, t$. The goal is to determine whether there exists a permutation $\sigma$ over $\{1, \ldots, t\}$ such that the $t$ vertices $v_{i,\sigma(i)}$, $i = 1, \ldots, t$, form a clique in $G$. If one puts vertices in a tabular of size $t \times t$, then we want to find a clique of $t$ vertices with exactly one vertex per row and one per column. This problem is trivially solvable in time $2^{O(t \log t)}$. Interestingly, (Lokshtanov et al., 2018) showed the following, where ETH stands for Exponential Time Hypothesis:

**Proposition 5.1.** *(Lokshtanov et al., 2018) Under ETH, Permutation Clique is not solvable in time $2^{o(t \log t)}$.*

We show that this problem remains hard even when being $O(\log t)$-position enhanced. Formally, the problem is defined as follows. We are given an instance $G = (V, E)$ of Permutation Clique on $t^2$ vertices, and we want to determine if there is a permutation $\sigma$ over $\{1, \ldots, t\}$ such that:

(1) $|\sigma(i) - i| \leq c \log t$ for all $i = 1, \ldots, t$;

(2) $\{v_{i,\sigma(i)} : i = 1, \ldots, t\}$ is a clique in $G$.

**Proposition 5.2.** *Under ETH, $(c \log t)$-Positioned-Enhanced Permutation Clique is not solvable in polynomial time (for any $c > 0$).*

*Proof.* Let us consider an instance $G = (V, E)$ of Permutation Clique, on $n = t^2$ vertices $\{v_{i,j} : i, j = 1, \ldots, t\}$. Let $t' = 2^{t/c}$. We build from $G$ a graph $G' = (V', E')$ on $n' = t'^2$ vertices $\{v'_{i,j} : i, j = 1, \ldots, t'\}$ where:

- For $i_1, i_2, j_1, j_2 \leq t$: $v'_{i_1,j_1}$ is adjacent to $v'_{i_2,j_2}$ if and only if $v_{i_1,j_1}$ is adjacent to $v_{i_2,j_2}$ in $G$;

- For $i, j \leq t$ and $\ell > t$: $v'_{i,j}$ is adjacent to $v_{\ell,\ell}$;

- For $\ell_1, \ell_2 > t$: $v_{\ell_1,\ell_1}$ is adjacent to $v_{\ell_2,\ell_2}$.

By construction, a (non trivial) clique in $G'$ is composed by a clique in $G$ plus 'diagonal' vertices $v_{\ell,\ell}$. Hence, there is a permutation clique of size $t'$ in $G'$ if and only if there is a permutation clique of size $t$ in $G$. Moreover, we know by construction that if there is a permutation clique induced by $\sigma'$ in $G'$, then for $\ell > t$ $\sigma(\ell) = \ell$ (diagonal vertices), and for $\ell \leq t$ $\sigma(\ell) \leq t$. Then in any case, $|\sigma(\ell) - \ell| \leq t = c \log t'$.

Suppose that we answer to $(c \log t')$-Positioned-Enhanced Permutation Clique in polynomial time $O(n'^c)$. The construction of $G'$ takes time $O(n')$, so we can solve Permutation Clique in time $O(n'^c) = O(t'^c) = 2^{O(t)}$. This is in contradiction with ETH. □

### 5.2. Hardness results for worse than $O(\log n)$-position-enhancement

Let us now go back to decomposable and $c$-local problems, and show that the $O(\log n)$-position enhancement is necessary for some problems.

**Proposition 5.3.** *For any unbounded increasing function $f$, $f(n) \log n$-Positioned-Enhanced Max Acyclic Subgraph is not solvable in polynomial time under ETH.*

*Proof.* We make a reduction from Max Acyclic Subgraph, which is not solvable in time $2^{o(n)}$ under ETH (from the linear reduction from vertex cover (Karp, 1972) and the hardness of vertex cover (Impagliazzo et al., 2001)). Let $G = (V, E)$ be a directed graph. We build a graph $G'$ by adding to $G$ dummy vertices so that $G'$ has $N = 2^{n/f(n)}$ vertices. We keep the arcs that are in $G$ but do not add any new arcs. Then the order in which we put dummy vertices does not change the objective function. So we know that there is an optimal solution such that $\sigma(i) = i$ for $i > n$ and $\sigma(i) \leq n$ for $i \leq n$, and there $\sigma$ restricted to $\{1, \ldots, n\}$ is an optimal solution for $G$. So we know that $|\sigma(i) - i| \leq n = f(n) \log N \leq f(N) \log N$.

Now, suppose that we can solve $f(n) \log n$-Positioned-Enhanced Max Acyclic Subgraph in polynomial time $O(N^c)$. Then we would solve Max Acyclic Subgraph in time $O(2^{cn/f(n)}) = 2^{o(n)}$ as $f$ is unbounded. This is impossible under ETH. □

**Proposition 5.4.** *For any unbounded increasing function $f$, $f(n) \log n$-Positioned-Enhanced TSP is not solvable in polynomial time under ETH.*

*Proof.* The proof is similar to the one of Proposition 5.3. This time, we add dummy vertices that are all at distance 0 from a given initial vertex $v$ (and all distances between them are 0), so that there are $N = 2^{n/f(n)}$ vertices in total. Then we know that there exists an optimal solution in which these vertices are ranked last in the permutation (the solution starts with $v$), so we get as previously $f(N) \log N$-position enhancement "for free". We get the same conclusion, using the fact that TSP is not solvable in time $2^{o(n)}$ under ETH (Impagliazzo et al., 2001). □

Similar proofs can be easily derived for other problems, such as Min Linear Arrangement and the single-machine sum of completion time problem considered in Section 3.

## Acknowledgement

Dimitris Fotakis has been partially supported by project MIS 5154714 of the National Recovery and Resilience Plan Greece 2.0 funded by the European Union under the NextGenerationEU Program. Michalis Xefteris was supported by the framework of H.F.R.I call "Basic research Financing (Horizontal support of all Sciences)" under the National Recovery and Resilience Plan "Greece 2.0" funded by the European Union – NextGenerationEU (H.F.R.I. Project Number: 15877). Panagiotis Patsilinakos has received support under the program "Investissements d'Avenir" launched by the French Government, project PSL JRC 2024 - N 2024-488.

## Impact Statement

This paper presents work whose goal is to advance the field of Machine Learning by further exploring its interactions with Theoretical Computer Science. There are many potential societal consequences of our work, none which we feel must be specifically highlighted here.

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
