# OpenReview forum: "Polynomial Time Learning Augmented Algorithms for NP-hard Permutation Problems"
_ICML.cc/2025/Conference — ICML 2025 poster_

### Official Review · Reviewer_aLF5 · 2025-02-20

**Overall Recommendation:** 2

**Summary:**

The paper considered permutation-based NP-hard problems with learning-augmented oracles. The paper proposed a novel framework that given a learning-augmented oracle on the ordering of pairs with correct probability $½+\Omega(1)$, solves certain families of NP-hard problems exactly in polynomial time and O(n log n) queries to the oracle predictions. The families of problems should satisfy either the ‘decomposable’ property, which means the ordering outside a subproblem does not matter for the subproblem itself, or the ‘locality’ property, which means the objective on a local permutation index is only dependent on the c neighboring indices. The paper provided an analysis (theorem 1.1) for the performance guarantees based on sorting without resampling results in BM [SODA’08]. The paper then studied a handle of example problems, including problems that satisfy the decomposable properties, e.g., maximum acyclic subgraph and minimum linear agreement, and the problems with the locality property, e.g. the traveling salesperson problem.

**Claims And Evidence:**

Yes. All theoretical claims are supported by proofs.

**Essential References Not Discussed:**

For related work, the paper mostly only discussed clustering and graph algorithms in the context of learning-augmented algorithms. A broader discussion about learning-related algorithms would be better (e.g., learning augmented frequency estimation, e.g., Aamand et al. [NeurIPS’23]; TSP problem, e.g., Chawla and Christou [APPROX’24]), especially given that some of these problems are closer to the permutation-based problems studied in this paper. This however is not a dealbreaker type of issue.

**Experimental Designs Or Analyses:**

N/A. Theoretical paper, no experiments are included.

**Methods And Evaluation Criteria:**

N/A. This paper does not contain experiments.

**Other Comments Or Suggestions:**

See my comments on 'strengths and weaknesses'.

**Other Strengths And Weaknesses:**

I have a mixed feeling about the paper. On the one hand, I like the idea of using only a sublinear number of predictions to obtain exact solutions for NP-hard problems in polynomial time. On the other hand, I think the properties of decomposable and locality are still restrictive. In particular, to motivate the study of NP-hard problems with predictions, the paper discussed some recent work on learning-motivated graph algorithms (e.g., CDGLP [NeurIPS’24]; BDSW [APPROX’24]); however, I believe none of these problems (max-cut, MIS, etc.) could be solved with the framework proposed by the paper.

The presentation of the paper also has much to be desired. The high-level description of the decomposable property is a bit too informal, and for a moment, I thought the framework could solve problems like max-cut as well (you can denote the cut as a permutation and find the actual cut from the permutation with $n$ extra time). It turned out that the property actually requires the overall problem to have local invariant properties as required by dynamic programming: this is not clear from the description. Also, I think the notation $\sigma(i)$ is overloaded between Definition 2.1 and Theorem 2.5. I believe for most places, when you write $\sigma(i)$, you mean the $i$-th element in the permutation $\sigma$. However, this is very confusing since it could also mean the permutation index of element $i\in [n]$. If I’m not mistaken, BM [SODA’08] used the notation in the latter sense.

**Questions For Authors:**

Most of the questions are mentioned in my criticism. One additional question: do you see more NP-hard graph problems that could be solved with your framework? This might help improve connections with your motivation.

**Relation To Broader Scientific Literature:**

The paper explored polynomial time algorithms for NP-hard problems with learning-augmented oracles. This area is important for beyond-worst-case algorithms and complexity.

**Theoretical Claims:**

On a high level, the main theorem of the paper follows from a ‘positional’ property in the algorithm of BM [SODA’08]. In particular, let $\pi^*$ be the permutation that maximizes the total number of queries consistent with the oracle and $\sigma^*$ be the optimal permutation. The algorithm of BM [SODA’08] satisfies the property that for each element $i\in [n]$, the distance between $\pi^*(i)$ and $\sigma^*(i)$ is at most $O(\log{n})$. Therefore, for each element on $\sigma^*(i)$, we could enumerate all possible $2^{O(\log{n})}$ ways to select locally from $\pi^*$ based on the decomposable or locality properties. This leads to the proof of Theorem 1.1.

I also checked the proofs for the maximum acyclic subgraphs, the minimum linear arrangement, and the TSP problems. These proofs look good as far as I could see.

---

> ### Author Rebuttal · Authors · 2025-03-28
>
> We thank the reviewer for the feedback and the provided questions.
>
> (Question/Comment) "A broader discussion about learning-related algorithms would be better (e.g., learning augmented frequency estimation, e.g., Aamand et al. [NeurIPS’23]; TSP problem, e.g., Chawla and Christou [APPROX’24]) [...]"
>
> (A) We thank the reviewer for this suggestion, we will provide a broader discussion about learning-augmented algorithms and add the references in the article.
>
> We note that Chawla and Christou [APPROX’24] do study a variant of TSP but in a very different model: they consider an online problem where the predictions are on the requests that will occur (prediction on locations and time Windows). On the other hand, Aamand et al.[NeurIPS’23] give algorithms with and without predictions that estimate frequencies of elements appearing in a data stream.
>
> (Q/C) About other graph problems that could be fit into our framework.
>
> (A) In the paper we mention Max Acyclic Subgraph, Min Linear Arrangement and TSP. The result for TSP easily generalizes to similar problems where the objective function is a sum on distances between adjacent vertices in the permutation (such as Hamiltonian Path, asymmetric cases, or even TSP with precedence constraints by incorporating these constraints in the DP).
> We provide here two other examples of NP-hard graph problems that can be handled in our Framework up to a slight adaptation:
>
> - Minimum traveling repairman, where given an edge-weighted graph we want to find a permutation $(\pi(1),...,\pi(n))$ of vertices which minimizes $\sum_{i>1} d_{\pi}(\pi(1),\pi(i))$ where $d_{\pi}(\pi(1),\pi(i))=\sum_{j=1}^{i-1}d(\pi(j),\pi(j+1))$ is the distance traveled by the traveller in the permutation from the first vertex $\pi(1)$ to vertex $\pi(i)$. Stated like this, it is not $c$-local (as it involves distances in the permutation from the initial vertex). However, this function also writes $\sum_{i\geq 1} (n-i)d(\pi(i),\pi(i+1))$. Then we can apply the same algorithm as we do for 1-local problems.
>
> - Min Cut Linear arrangement, where given a graph we want to find a permutation on vertices which minimizes $\max_{q=1,...,n} f_{\pi}(q)$, where $f_{\pi}(q)=| ( u,v ) \in E: \pi^{-1}(u)<q<\pi^{-1}(v)|$ counts the number of edges "cut" at integer point $q$. The objective function to be minimized is a max, not a sum. However, we can still apply our framework with a slight generalization of the definition of decomposition (basically, we need a max instead of a plus, with the condition that there is an optimal permutation on $[i,j]$ that is made of optimal subpermutations on $[i,s]$ and $[s+1,j]$, which is the case here).
>
>
> (Q/C) "the paper discussed some recent work on learning-motivated graph algorithms (e.g., CDGLP [NeurIPS’24]; BDSW [APPROX’24]); however, I believe none of these problems (max-cut, MIS, etc.) could be solved with the framework proposed by the paper."
>
> (A) Indeed, the point is that problems like Max Cut or MIS are not permutation problems, so they do not fall into our framework. Let us make a remark though. Suppose that for a problem like Min Bisection (or Max Cut) we ask for a permutation, having in mind that the first $n/2$ elements are on one part of the bisection and the last $n/2$ are on the other part. Then our framework easily applies, and we can solve these problems in polytime ... provided that we have the required oracle on queries. But the point is that the queries are not natural for these problems (asking whether $v$ is before $u$ or not in the "permutation"); moreover the answers cannot be deduced from the natural queries "is $u$ on the left or on the right part of the bisection" (we would not get independent answers).

---

> > ### Comment · Reviewer_aLF5 · 2025-04-02
> >
> > Thanks for the clarification and the additional example. I agree that your framework would apply as long as there is a 'natural' permutation oracle. I also appreciate the responses on why you cannot solve max-cut and the case when how an oracle for max-cut would be in your framework (and I agree it won't be natural).
> >
> > Since the paper did give me a hard time understanding, I think the exposition should be significantly improved to incorporate the clarifications in the rebuttal. In light of this, I'll keep my score as it is. However, **I won't stand in the way of the paper being accepted**. I do hope the authors clarify the points in the rebuttal if this paper gets accepted.

---

> > > ### Author Response · Authors · 2025-04-07
> > >
> > > We will indeed make the changes to clarify these points if the article gets accepted (and anyway in a revised version).

---

### Official Review · Reviewer_YZRG · 2025-02-27

**Overall Recommendation:** 4

**Summary:**

The paper presents a framework for studying NP hard optimization problems in the framework of learning-augmented algorithms. Specifically, the authors consider a class of permutation problems, where a solution is determined by a total order of a set of $n$ items, such as TSP, single machine scheduling, etc, and that satisfy certain properties such a c-locailty or decomposition. An algorithm has access to a noisy (consistent) oracle that outputs for two elements $u,v$ whether $u$ comes before $v$ in a fixed optimal solution with probability of at least $\frac{1}{2}+\varepsilon$ with $\varepsilon > 0$.

They present a polynomial-time algorithm for this class of permutation problems that outputs an optimal solution with high probability. Moreover, their algorithm only uses $O(n \log n)$ queries to the noisy oracle, hence can be considered to be parsimonious. Finally, they show that many well-known NP-hard permutation problems lie in this class, and present tightness of their bounds and also of their approach.

Their approach builds up on a beautiful result by Braverman & Mossel, who considered the problem of sorting using a noisy oracle without resampling. On a high-level, they first use this result to compute a permutation $\pi^*$ that minimizes the total number of inversions compared to the oracle. Then they use a property proven by Braverman & Mossel that such a permutation is "close" to an optimal solution. They show the c-locality properties or the decomposition properties suffice to compute the optimal solution in polynomial time given $\pi^*$ .

**Claims And Evidence:**

All claims are proven formally.

**Essential References Not Discussed:**

As far as I see, all relevant references are there.

**Experimental Designs Or Analyses:**

N/A

**Methods And Evaluation Criteria:**

N/A

**Other Comments Or Suggestions:**

I find the definition of $s$-optimal a bit confusing, as it seems that the $s$ will be user interchangeably, which however does not happen in the paper. Thus, I would suggest to use a different term for this property.

**Other Strengths And Weaknesses:**

Strengths
- The result is interesting and increases our understanding on $\varepsilon$-accurate predictions.
- The approach reuses previous results, and thus, is quite simple and easy to present and hence understandable
- The result captures many different problems by definition an appropriate subclass of permutation problems.

Weaknesses
- The model of $\varepsilon$-accurate predictions has previously been used for other problems. Thus, one can argue that the present paper only uses known techniques.
- The results heavily depends on the result of Braverman & Mossel and thus slightly lacks technical novelty. Given that this is an ML conference, I would not weigh this point too much.

In summary, I believe that this is a neat and approachable result that contributes well to the field of algorithms with predictions.

**Questions For Authors:**

None

**Relation To Broader Scientific Literature:**

The paper falls in a line within the area of learning-augmented algorithms that studies the power of predictions for classic questions in complexity theory. The model there an algorithm receives a "correct" signal with probability strictly larger than 1/2 has previously been successfully applied to break the hardness of other problems with high probability.

The paper also relates their algorithm to the idea of "warm starting" algorithm, a technique that previously has been used for improving the runtime with predictions.

**Theoretical Claims:**

I checked a few arguments in Section 3 and the proof in Section 4.

---

> ### Author Rebuttal · Authors · 2025-03-28
>
> We would like to thank the reviewer for the feedback.
>
> About the comments of the reviewer: The model of epsilon-accurate predictions has indeed been used for other problems. So, the model is not novel, but we would like to note that our work is completely different from previous works on this model both on the techniques used, on the obtained results (polytime exact algorithms and not approximation algorithms as previously studied) and on the problems tackled (permutation problems).

---

### Official Review · Reviewer_bUbX · 2025-03-10

**Overall Recommendation:** 4

**Summary:**

The paper studies how to tackle NP-hard permutation problems with the help of predictions on the relative position of any two elements with respect to some fixed unknown optimal permutation. Each query is independently correct with probability at least $1/2 + \varepsilon$ for some unknown $\varepsilon$. Building upon a prior result [BM08] that the "most consistent" permutation with respect to these faulty predictions is O(log n)-"close" to the fixed unknown optimal permutation, the authors propose two sufficient conditions (decomposition and c-locality) under which a "close" permutation can be used as a warm-start to optimally solve the NP-hard problem in polynomial time using dynamic programming methods. The authors further prove hardness results that their results for decomposition and c-locality are tight, and that having a "close" permutation does not always suffice in general.

[BM08] Braverman, M. and Mossel, E. Noisy sorting without resampling. In Proc. 19th Symp. Discret. Algorithms (SODA), pp. 268–276, 2008.

**Claims And Evidence:**

Yes, proofs were given.

**Essential References Not Discussed:**

Not that I am aware of

**Ethical Review Concerns:**

NIL

**Experimental Designs Or Analyses:**

No experiments as it is a theoretical paper.

**Methods And Evaluation Criteria:**

Method is sound.

**Other Comments Or Suggestions:**

- Suggestion: Mention that the predictions are with respect to some unknown fixed optimal solution in the abstract? My initial reading of the abstract immediately raised questions in my head about globally inconsistent predictions but was only resolved when I read Line 62.
- Typo: Is there an extra space before the full stop at the end of Theorem 1.1?
- Suggestion: There were a few places where $\pi^*$ was described as "optimal solution" instead of "s-optimal solution". Maybe add in "s-" so that it avoids confusion with the underlying unknown optimal permutation $\sigma^*$?
- Paraphrase suggestion for the first sentence paragraph before Section 3: "Motivated by Lemma 2.6, we give two sufficient conditions ..., for k = O(log n), in the next section." I personally found the phrasing "Then, we mainly exhibit..." a bit awkward / does not flow well into the next section, but this is just a minor rephrasing suggestion that you may choose to ignore.
- Typo: In Definition 3.1, I believe the $S(\cdot)$ should be $S_{\sigma}(\cdot)$? There are such instances $S(\cdot)$ in the rest of this document. Also, is $\mathcal{S}$ in the proof of Lemma 4.1 the same thing as $S_{\sigma}(\cdot)$? Maybe do a ctrl-F on "S(" and "\mathcal{S}(" and check?
- Typo: Inconsistent bracketing on the last sentence of proof in Lemma 4.1
- Right side of Line 371 in the proof of Lemma 4.2: "i+1, i+c in $\sigma$" should be "i+1 and i+c in $\sigma$".

**Other Strengths And Weaknesses:**

# Strengths
- The authors provide two very general conditions which extends the applicability of [BM08]
- Hardness results are given to show the tightness of their results
- The paper is very well-written in general; the exposition and proofs are very easy to follow
# Weaknesses
- While theoretically interesting, it is unclear to me how such a prediction can be available in practice for each of the problems mentioned in the paper

[BM08] Braverman, M. and Mossel, E. Noisy sorting without resampling. In Proc. 19th Symp. Discret. Algorithms (SODA), pp. 268–276, 2008.

**Questions For Authors:**

- Question: Technically speaking, $\varepsilon$ needs to be constant otherwise the running time $n^{O(-\varepsilon^4)}$ is not polynomial, right? I can't remember if "constant $\varepsilon$" was mentioned somewhere in the paper.
- For completeness of the problem modeling, please provide references for NP-hardness of minimum linear arrangement and TSP.
- In Lemma 4.1, what is the purpose of introducing $n^d$? $d$ was never mentioned in the proof and we actually need $d$ to be constant otherwise $n^d \not\in poly(n)$, right? How about changing the statement to $O(n \cdot 2^{O(k)} \cdot t)$ where $t \in poly(n)$ is the time to compute the value of a solution? Also, please mention where the dependency on this computation time shows up in the proof. Currently it is not being mentioned at all.

**Relation To Broader Scientific Literature:**

The work here contributes to the field of learning-augmented algorithms in the form of speeding up solving of NP-hard problems in the presence of noisy predictors. More precisely, the authors provide two very general conditions (decomposability and c-locality) which extends the applicability of [BM08].

[BM08] Braverman, M. and Mossel, E. Noisy sorting without resampling. In Proc. 19th Symp. Discret. Algorithms (SODA), pp. 268–276, 2008.

**Theoretical Claims:**

I checked the proofs. Looks correct.

---

> ### Author Rebuttal · Authors · 2025-03-31
>
> We thank the reviewer for the feedback and the provided questions. We will implement the suggestions on a revised version.
>
> - (Comment) While theoretically interesting, it is unclear to me how such a prediction can be available in practice for each of the problems mentioned in the paper [BM08]
>
> (A) This point is also raised by Reviewer PUBJ ("Can the authors comment on how easy the ranking permutations would be learn / initialize in practice? The question of where the predictions are coming from (just the motivations even) is not really discussed."). We provide here a common answer to these points about the predictions.
>
> Our framework and results indeed ask for good predictions of optimal solutions for hard optimization problems. This is clearly a strong assumption. The soundness bound of $1/2+\epsilon$ is actually not very strong (in the sense that any $\epsilon>0$ works), but the independency might be harder to get in practice. We do not claim that there are currently ML algorithms on the shelf that have the required accuracy.  However, we stress the fact that a large number of recent works do focus on getting predictions for discrete optimization problems. This area, part of the larger question of using ML for optimization, is still in its early stage of development but is growing very fast. We refer to [1,2] for survey-like papers with many references in it. We also quote a specific work on predictions for problems including (Euclidean) TSP [3]. Hence, we believe that the quick development of the field makes it reasonable to hope for some accurate predictors for some permutation problems  in the near future, making our framework possibly relevant also from a more practical viewpoint. Also, we think that the theoretical result we give on the potential fruitful use of good predictions in solving hard permutation problems might be a further motivation for studying, in ML, predictions for solutions of optimization problems.
>
>
> Nevertheless, there are restricted settings where one could outline how such noisy predictions of the optimal permutation can be learned. E.g., consider a setting where instances are stationary and instance stationarity implies that the optimal solution / permutation of each instance is a noisy sample from a Mallows distribution $\mathcal{M}(\pi^\ast, \beta)$ with $\pi^\ast$ corresponding to the optimal permutation of the "mean" instance and the noise parameter $\beta$ accounts for the variance of the instance distribution (for a more concrete example, we may think of a Euclidean TSP setting where the location of each vertex $v_i$ is drawn from a fixed distribution $\mathcal{V}_i$ and the distributions $\mathcal{V}_1, \mathcal{V}_2, \ldots, \mathcal{V}_n$ are mutually independent). Such a Mallows distribution $\mathcal{M}(\pi^\ast, \beta)$ can be learned [4] up to KL distance $\epsilon$ from the optimal (or near-optimal) solutions of $O(\frac{\ln(1/\delta)}{n \epsilon^2 } + \frac{\ln(n / \delta)}{\beta})$ instances sampled independently from the instance distribution (where $n$ is the number of vertices and $\delta$ is the probability that the learning algorithm fails). Samples from $\mathcal{M}(\pi^\ast, \beta)$ can be used as predictions (in the form of noisy rankings, as discussed in the response to Reviewer PUBJ) in our approach.
>
> [1] Y. Bengio, A. Lodi, A. Prouvost. Machine learning for combinatorial optimization: A methodological tour d’horizon. EJOR 290, Issue 2, 16 April 2021, Pages 405-421
>
> [2] Q. Cappart, D. Chételat, E. B. Khalil, A. Lodi, C. Morris, P. Velickovic. Combinatorial Optimization and Reasoning with Graph Neural Networks. J. Mach. Learn. Res. 24: 130:1-130:61 (2023)
>
> [3] E. B. Khalil, H. Dai, Y. Zhang, B. Dilkina, L. Song: Learning Combinatorial Optimization Algorithms over Graphs. NIPS 2017: 6348-6358
>
> [4] R. Busa-Fekete, D. Fotakis, B. Szörényi, M. Zampetakis. Optimal Learning of Mallows Block Model. COLT 2019: 529-532.
>
> - (Question) Technically speaking, needs to be constant otherwise the running time $n^{O(1/\epsilon^4)}$ is not polynomial, right?
>
> (A) Indeed, this is needed for the running time to be polynomial. We will make this explicit.
>
> - (Question) For completeness of the problem modeling, please provide references for NP-hardness of minimum linear arrangement and TSP.
>
> (A) Indeed, we will do this.
>
> - (Question) In Lemma 4.1, what is the purpose of introducing $n^d$? $d$ was never mentioned in the proof and we actually need to be constant otherwise , right? How about changing the statement to $O(n.2^{O(k)}t)$ where $t$ is the time to compute the value of a solution? Also, [...] being mentioned at all.
>
> (A) Indeed, this reformulation $O(n.2^{O(k)}t)$ is correct and simpler, we will make the modification. The dependency on the time to compute the value of the solution arises when selecting the best partition of $I'(i,j)$ into two sets among $2^{O(k)}$ possible choices. We will add a remark about this in the proof.

---

> > ### Comment · Reviewer_bUbX · 2025-04-02
> >
> > Thank you very much for the response and clarifications! I intend to maintain my score.
> >
> > Please make the changes you mentioned here and add the discussion written here for me and Reviewer PUBJ in your revision. Thanks!

---

> > > ### Author Response · Authors · 2025-04-07
> > >
> > > We will indeed make these changes and add the discussion if the article gets accepted (and anyway in a revised version).

---

### Official Review · Reviewer_PUBJ · 2025-03-17

**Overall Recommendation:** 4

**Summary:**

This paper studies a broad class of optimization problems that can be captured via permutations with the help of predictions. For example consider the TSP problem: the optimal solution corresponds to a permutation of the input points. The predictions are weather point i precedes point j in the permutation and each pair of information is correct with probability 1/2 + eps iid (note one cannot re-query the same pair). In generally, the authors note that many classic optimization problems can fit this framework.

At a high level, the idea is to draw connection to a prior studied problem of sorting with such similar noisy information. Using ideas from a series of papers from Braverman and Mossel, the submission at a high level, first reconstructs a permutation that matches the information given by the prediction. There is a key property that every point won't be 'too far away' (within O(log n)) of its true position in the reconstructed permutation. If the optimization problem has the special structure that one can ignore such far away interactions (the authors identify many such examples), one can then optimally fix the local mistakes by using dynamic programming to get an overall polynomial time. Another nice aspect is that the total number of queries made to the oracle is bounded by O(n log n) but this seems to follow from the Braverman Mossel results.

In summary, I am quite positive of the paper for the following reasons:

There are several aspects of this paper that I enjoy:
- Studies NP hard optimization problems: this seems like a promising area for learning augmented algorithms (and was the main motivation of this paper following some nice work on max cut)
- Introduces a general framework by looking at problems that can be represented as permutations.
- The algorithms are very simple and intuitive

However, here are some cons:
- Since the algorithms are quite simple, i find the lack of experiments to be a bit disappointing since i think they could work very well.
- The question of where the predictions are coming from (just the motivations even)  is not really discussed
- I could not easily separate the technical contributions made by the paper and the works of Braverman and Mossel. I recommend better highlighting what arguments are novel to the submission. However, I view a major part of the submission as a conceptual contribution so this point is somewhat minor to me.

**Claims And Evidence:**

Yes all claims seem sound.

**Essential References Not Discussed:**

Minor comment but [1] is also among the first paper to use predictions to warm start and optimization problem and then solve the initialized problem optimally.

[1] Faster Fundamental Graph Algorithms via Learned Predictions. ICML 22

**Experimental Designs Or Analyses:**

N/A

**Methods And Evaluation Criteria:**

No empirical evaluation.

**Other Comments Or Suggestions:**

A minor point: I was really confused in the beginning where the dependence on epsilon goes to. For example, if epsilon = 0 then the advice is useless. Only after a few passes did I understand that the running time blows up (so the running time is polynomial for any fixed epsilon). This makes sense but the authors should highlight this a bit more.

**Other Strengths And Weaknesses:**

None

**Questions For Authors:**

- What if you are given a 'noisy' ranking in the beginning? This has some pros and cons: pros is the information is only O~(n) bits rather than n^2 (but harder to incorporate 1/2 + eps criteria).
- Are there lower bounds stating that Omega(n log n) queries are necessary?
- Can the authors comment on how easy the ranking permutations would be learn / initialize in practice?

**Relation To Broader Scientific Literature:**

The paper extend a new direction in learning-augmented algorithms which studies NP hard problems by introducing a framework for solving problems related to permutations.

**Theoretical Claims:**

I checked some theoretical proofs and they seem sound.

---

> ### Author Rebuttal · Authors · 2025-03-31
>
> We would like to thank the reviewer for the feedback and the provided questions.
>
> - (Comment) Minor comment but [1] is also among the first paper to use predictions to warm start ...
>
> (A) Thank you, we will add this reference on a revised version.
>
> - (Comment) I was really confused in the beginning where the dependence on epsilon goes to. For example, if epsilon = 0 then the advice is useless. [...]
>
> (A) Indeed, this is also pointed by another reviewer, we will make the dependency on epsilon more explicit in a revised version.
>
> - (Question) What if you are given a 'noisy' ranking in the beginning? This has some pros and cons: pros is the information is only O~(n) bits rather than n^2 (but harder to incorporate 1/2 + eps criteria).
>
> (A) First, we underline that our approach also uses O(n log n) bits of predicted information. If we understand correctly, the question is "what happens if we have predictions queries relative to a noisy ranking sigma' (sigma' being noisy wrt to the true optimal ranking sigma^*)?".
>
> It depends of course on the noise. An easy case is when |sigma'(i)-sigma^*(i)|=O(log n) for all i (whp), as then the results are still valid since we still have k-enhanced problems with k=O(log n). If this 'pointwise' bound does not occur, it is not clear whether weaker assumptions on the noise could be sufficient.
>
> An interesting case where this bound occurs is when the prediction is a noisy permutation sampled from a Mallows distribution [1]. A Mallows distribution is defined w.r.t. the optimal permutation $\pi^*$ and a noise parameter $\beta \in [0, 1]$. Then, the probability of generating a permutation $\sigma$ is proportional to $e^{- \beta d_{KT}(\sigma, \pi^\ast)}$, where $d_{KT}(\sigma, \pi^\ast)$ is the Kendal tau distance between $\pi^*$ and $\sigma$ (the Kendal tau distance accounts for the number of pairwise disagreements between $\pi^*$ and $\sigma$). The case where $\beta = 0$ corresponds to the uniform distribution, while the case where $\beta \leftarrow \infty$ allocates probability $1$ to $\pi^\ast$. This case was also studied in the follow-up paper by Braverman and Mossel [1]. It can be shown (e.g., [1, Lemma 17]) that for any constant $\beta > 0$, $|\sigma(i)-\pi^*(i)| = O(\log n)$ for all $i$ (whp). Therefore, our results can be extended to the case where the prediction is provided in the form of a noisy ranking sampled from a Mallows distribution with the optimal permutation as $\pi^\ast$ and a constant noise parameter $\beta > 0$ (here $\beta$ plays the role of $\epsilon$ and our running time has $\mathrm{poly(1/\beta)}$ in the exponent).
>
> [1] Braverman, M. and Mossel, E. Sorting from noisy information, 2009. arXiv:0910.1191.
>
> [2] C. L. Mallows. Non-null ranking models. Biometrika, 44:114–130, 1957.
>
> - (Question) Are there lower bounds stating that Omega(n log n) queries are necessary?
>
> (A) Actually $\Omega(n \log n)$ queries are necessary even in the case of perfect predictions. Indeed, when answers to queries are correct (with probability 1) the Noisy Sorting Without Resampling problem becomes sorting by comparisons (where comparisons are queries). It is well known that sorting by comparisons needs $\Omega(n \log n)$ comparisons (even for randomized algorithms), so even in the case of 'perfect predictions' (correct comparisons)  $\Omega(n \log n)$ queries are necessary to sort/recover the order.
>
> Thanks for pointing this, we will add this remark on a revised version.
>
> - (Question and Comment) Can the authors comment on how easy the ranking permutations would be learn / initialize in practice? The question of where the predictions are coming from (just the motivations even) is not really discussed.
>
> (A) This point is also raised by reviewer bUbX. Please read our answer in the rebuttal of reviewer bUbX (first point (comment) in that rebuttal), thanks.

---

### Decision · Program_Chairs · 2025-05-01

**Decision:**

Accept (poster)

**Comment:**

The paper introduces a framework for applying learning-augmented algorithms to NP-hard permutation problems such as TSP and single-machine scheduling, where solutions rely on item orderings, and explores how algorithms can leverage a noisy but consistent oracle to guide decisions under structural properties.

It seems many of the same concerns are shared across reviewers regarding:
- the specific technical contributions of the paper beyond the observation of applying the Braverman-Mossel results
- discussion of the motivation or ability to acquire predictions in the suggested oracle model

Additionally, though the paper studies NP-hard problems with advice, it is not clear that these NP-hard problems are unified and central to ML audiences. Although the initial review scores are positive, subsequent reviewer discussion did not reach a resolution on these issues, reflecting a more borderline decision.